

# DACCOR–Detection, characterization, and reconstruction of repetitive regions in bacterial genomes

Alexander Seitz, Friederike Hanssen and Kay Nieselt

Center for Bioinformatics (ZBIT), Integrative Transcriptomics, Eberhard-Karls-Universität Tübingen, Tübingen, Germany

## ABSTRACT

The reconstruction of genomes using mapping-based approaches with short reads experiences difficulties when resolving repetitive regions. These repetitive regions in genomes result in low mapping qualities of the respective reads, which in turn lead to many unresolved bases. Currently, the reconstruction of these regions is often based on modified references in which the repetitive regions are masked. However, for many references, such masked genomes are not available or are based on repetitive regions of other genomes. Our idea is to identify repetitive regions in the reference genome de novo. These regions can then be used to reconstruct them separately using short read sequencing data. Afterward, the reconstructed repetitive sequence can be inserted into the reconstructed genome. We present the program detection, characterization, and reconstruction of repetitive regions, which performs these steps automatically. Our results show an increased base pair resolution of the repetitive regions in the reconstruction of *Treponema pallidum* samples, resulting in fewer unresolved bases.

## INTRODUCTION

The reconstruction of a genome from sequencing reads can be achieved using a de novo assembly, where overlaps of the reads are identified and thus extended into longer continuous sequences called contigs. Alternatively, if a closely related reference genome is available, the reads can be mapped against this genome, where the reconstruction is based on the consensus of the reads for each base.

For the mapping-based approach, programs such as such as `BWA` (*Li & Durbin, 2009*) or `Bowtie2` (*Langmead & Salzberg, 2012*) are used to align short reads generated by Next-Generation-Sequencing (NGS) technologies to a known reference genome (*Veeramah & Hammer, 2014*). The consensus sequence of the aligned reads can then be used to generate the genomic sequence of the newly sequenced sample, assuming that the sample was sequenced with a sufficient coverage depth. This allows for the fast identification of short insertions, deletions, and single nucleotide variations (SNVs) or single-nucleotide polymorphism (SNP). The mapping programs typically calculate a score for each aligned read that corresponds to the quality of the alignment (*Li, Ruan & Durbin, 2008*). The score

Corresponding author
Alexander Seitz,
alexander.seitz@uni-tuebingen.de

quantifies the probability that a read is placed at the correct genomic position. Reads with a low mapping quality can be filtered out to remove reads that might stem from contaminations or were sequenced with low sequencing quality (*Smith, Xuan & Zhang, 2008*). Besides bad quality reads, also reads mapping to repetitive regions could yield low-quality scores if they cannot be mapped to a unique position. Filtering of reads with low mapping qualities would also include these reads. This filtering is often conducted in the context of ancient DNA (aDNA) (*Bos et al., 2016*), so that for such samples the repetitive regions of the respective reconstructed genomes are generally affected, resulting in many unknown (N) characters. Also, de novo assembly approaches have problems in reconstructing repetitive regions, at least those that are significantly longer than the read lengths (*Simpson & Durbin, 2012*).

However, repetitive regions play an essential role in many genomes (*Shapiro & von Sternberg, 2005*). Hundreds to thousands of such regions are present in prokaryotic and eukaryotic chromosomes (*Treangen et al., 2009*). The human genome, for example, consists of approximately 50% repetitive regions (*International Human Genome Sequencing Consortium, 2001*). Tandem repeat regions in bacteria appear to be associated with outer membrane proteins, which suggests that they help pathogens to adapt to their hosts (*Denoeud & Vergnaud, 2004*). In the case of the bacterium *Treponema pallidum*, repetitive sequences in the *arp* gene are used to distinguish between the subspecies that cause veneral syphilis (*T. pallidum pallidum*), nonveneral yaws (*T. pallidum pertenue*), and bejel (*T. pallidum endemicum*), which is not possible using serological tests (*Harper et al., 2008*).

New sequencing technologies, like the Illumina SLR platform, PacBio, or Oxford Nanopore, can create long reads that can span most repetitive regions to resolve them (*Huddleston et al., 2014*). However, it is not always possible to apply these technologies to DNA samples. For example, in aDNA projects, the average extracted fragment length is approximately 44–72 base pairs (*Sawyer et al., 2012*). Sequencing these fragments with long read technology would not result in any information gain, as even Illumina short read technologies can sequence the whole fragment. Short DNA fragments are not limited to aDNA. They also appear in the sequencing of hard to cultivate pathogens, like *T. pallidum* (*Arora et al., 2016*).

To better resolve the repetitive regions using short reads, researchers often first mask duplicated and low-complexity regions prior to read mapping (*Frith, Hamada & Horton, 2010*). One example would be the human reference genome, where a masked version is already available (*University of California Santa Cruz (UCSC), 2014*). If no masked genome is available, programs like `RepeatMasker` (*Smitt, Hubley & Green, 1996*) can identify these regions and create a masked reference. `RepeatMasker` uses libraries of known repetitive regions and compares them to the input sequence. While this allows masking of the genome, de novo identification of repetitive regions is not possible.

One program that can identify repetitive regions de novo is `VMatch` (*Kurtz, 2003*). It uses suffix-arrays (*Weiner, 1973*) to identify the repetitive regions. `VMatch` has been applied in multiple genome projects for annotation of repetitive regions (e.g., by *Lindow & Krogh (2005)*), as well as masking tasks (*Assuncao et al., 2010*). Of course, there are

other de novo repeat finding programs, like `RepeatExplorer` (*Novák et al., 2013*), a galaxy-based web tool, or `RepARK` (*Koch, Platzer & Downie, 2014*), which is based on the sequencing reads and not on the respective reference genome. So far, no tool exists that allows users to combine repeat identification and genome reconstruction from NGS data.

The general idea of our approach first starts with a de novo identification of all repetitive regions in a given reference genome. Reads from NGS data are then mapped against the reference genome as well as all identified repetitive regions individually, to reconstruct them. To increase the base pair resolution of the reconstructed genome for each sample, the individually reconstructed repetitive regions are then used to replace the corresponding reconstructed repetitive regions of the full genome. All of these steps are combined in a fully automatic tool which we call detection, characterization, and reconstruction of repetitive regions (`DACCOR`) in genomes.

## METHODS

Detection, characterization, and reconstruction of repetitive regions consists of three major steps (see central column of Fig. 1). It first conducts a de novo identification of all repetitive regions for a given reference genome. For each identified repetitive region and each NGS data sample, the respective sequence is then reconstructed using the mapping-based pipeline efficient ancient genome reconstruction (`EAGER`). For each sample, `EAGER` is also used to reconstruct a full genome. Finally, all repetitive regions in the reconstructed genome are replaced by the individually reconstructed repetitive sequences.

To create an integrated version to identify repetitive regions, our de novo approach uses a $k$-mer based approach, similar to `WindowMasker` (*Morgulis et al., 2005*). The workflow to identify these regions can be split up into six steps (see left part of Fig. 1). In the first step, the reference genome is divided into all its $k$-mers and all nonunique $k$-mers are stored. In the second step, matching $k$-mer pairs overlapping at $(k-1)$ positions are combined into $(k+1)$-mers and stored again. This second step is repeated until all maximal unique repetitive regions are identified. Afterward, low-complexity regions, e.g., long regions consisting only of the same base, are identified. This step is necessary, as the size of the currently identified repetitive regions is increased by one in each iteration (step 2). Thus, in the end, these regions are represented as two different repetitive regions, even though they are completely identical. This third step identifies and combines them to get the maximal repetitive regions.

In the following two steps, repetitive regions with mismatches are identified. Here, a mismatch-marker representing any unknown base is added to the end of all currently identified maximally exact repetitive regions. This representation allows us to combine two previously identified repetitive regions that are separated by these mismatches. If they only differ by one base after a mismatch marker is added to all sequences, they are now represented as the same sequence and can be combined. This can be repeated to allow for multiple mismatches. Matching pairs of separated regions can thus be combined into repetitive regions with mismatches. In the last filtering step, leading and trailing mismatch markers, as well as repetitive regions that do not fulfill the user-defined criteria

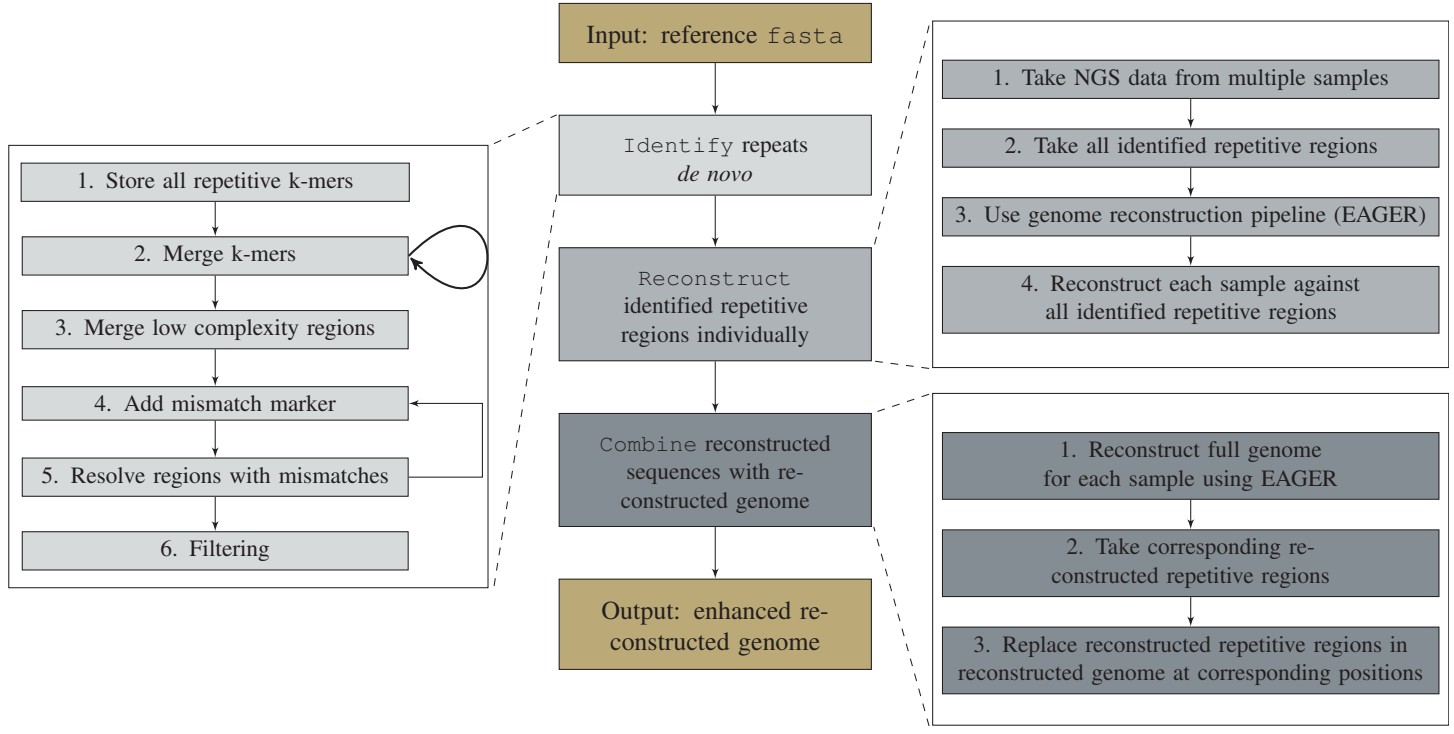

**Figure 1 Workflow of the repeat reconstruction pipeline `DACCOR`.** First, all repetitive regions in a given reference genome are identified. `DACCOR` comes with its own de novo repeat identification tool, which is separated into six steps. However, `DACCOR` is not limited to this approach and it can be exchanged with any other repeat identification tool. Afterward, each repetitive region is reconstructed separately using `EAGER`. Finally, the individually reconstructed regions are combined with the reconstruction of the full genome by replacing the corresponding base pairs in the reconstructed genome with the ones of the reconstructed repetitive regions. This leads to an increased base pair resolution of these regions.

(e.g., length of the repetitive regions), are removed. All remaining mismatch markers are replaced with the character *N* (see Fig. S1 for an example). However, this approach can only handle base pair mismatches but no insertions or deletions. If indels exist within the repetitive regions, they will be identified separately and are still used in the following steps to reconstruct them.

The methodology for identifying repetitive regions, as described above, is implemented in the `identify` subprogram of `DACCOR` and can be used separately for the de novo identification of repetitive regions in a given reference genome. The `identify` subprogram can identify repetitive regions within as well as between different chromosomes or a bacterial genome and its plasmids. This is done by combining the *k*-mers of all sequences in a given multi-`fasta` reference file. To be able to match identified repetitive regions to the corresponding sequence, a unique offset is added to the indices of the start location of each region.

Detection, characterization, and reconstruction of repetitive regions is not limited to our approach for the identification of repetitive regions. It fully supports the output format of `Vmatch`. For this, `Vmatch` needs to be run independently. Its result can then be used instead of our repeat finding method. The regions identified by `Vmatch` are then extracted and saved in separate `fasta` files to be used in the next step. All identified

repetitive regions are also stored in one multi-`fasta` file, in addition to the files containing each one repetitive region for the reconstruction. Furthermore, a summary of all identified repetitive regions, as well as a `bed` file to view the location of these regions in a genome viewer are created. Users can specify a minimum length for all reported repetitive regions.

To increase base pair resolution of genomes reconstructed from NGS data, the repetitive regions are used as separate references for the reconstruction of the individual repetitive regions in multiple NGS sequencing samples in addition to the reconstruction of the full genome. To correctly reconstruct both ends of each respective repetitive region, a flanking region of a user predefined length is added on both the 3′ and 5′ end of each region, so that reads overlapping only part of the respective region can be mapped correctly.

To reconstruct the repetitive region from NGS sequencing data, we use the `EAGER` pipeline developed by *Peltzer et al. (2016)* (version 1.92.37). In `DACCOR`, `EAGER` is used to preprocess the raw reads, including adapter clipping and quality trimming, to map them against a given reference (with `BWA` (*Li & Durbin, 2009*) version 0.7.17-r1188), and generate a consensus sequence in the `fasta` file format using `GATK` (*McKenna et al., 2010*) version 3.8-0-ge9d806836 followed by `VCF2Genome` (*Peltzer et al., 2016*) (version 0.91). For the consensus reconstruction, it uses the results of the genotyping results, following `GATK` Best Practice's guidelines (*Van der Auwera et al., 2013*).

The `reconstruct` subprogram of `DACCOR` automatically generates `EAGER` configuration files for a given reference, its identified repetitive regions, and multiple sequencing samples.

The `combine` subprogram uses the `EAGER` output of the individually reconstructed repetitive regions, as well as the reconstruction of the whole genome and combines them. Because the origin of the repetitive regions is known, its reconstructed sequences can replace the bases in the original full genome reconstruction. For each repetitive region, the respective positions in the genome are replaced if the original reconstruction resulted in an unknown (*N*) character.

To be able to automatically assemble a genome with all its repetitive regions, the subprogram `pipeline` connects the described subprograms `identify`, `reconstruct`, and `combine`.

To evaluate our method, we applied `DACCOR` to four bacterial genomes of various lengths and repetitiveness, namely *T. pallidum subsp. pallidum str. Nichols*, *Mycobacterium leprae TN*, *Escherichia coli str. K-12 substr. W3110*, and the genome of *Shigella flexneri 2a str. 301*, without its plasmids (see Table 1 for RefSeq IDs). We first compared the step of the identification of repetitive regions with `VMatch`, allowing for one mismatch in repetitive regions of a minimum length of 101 base pairs (the length of typical Illumina HiSeq reads). We then applied our proposed reconstruction method to 106 syphilis samples published by *Arora et al. (2016)* (70 samples), *Pinto et al. (2016)* (25 samples), *Sun et al. (2016)* (10 samples), as well as read data from the reference genome (see Table 1 for SRA project IDs) to reconstruct the sequences of the 16S and 23S rRNA, which are duplicated in the bacterium *T. pallidum*. For this, we first used `DACCOR` to identify all

**Table 1  Published reference genomes and NGS data used in this study.**

| Bacterium | RefSeq ID |
| --- | --- |
| *Treponema pallidum subsp. pallidum str. Nichols* | NC_021490.2 |
| *Mycobacterium leprae TN* | NC_002677.1 |
| *Escherichia coli str. K-12 substr. W3110* | NC_007779.1 |
| *Shigella flexneri 2a str. 301* | NC_004337.2 |
| **Publication** | **SRA Project ID** |
| *Arora et al. (2016)* | PRJNA313497 |
| *Pinto et al. (2016)* | PRJNA322283 |
| *Sun et al. (2016)* | PRJNA305961 |

repetitive regions with at most five mismatches and a minimal length of 101 base pairs in the *T. pallidum* genome. We also reconstructed the full genomes of the samples using the standard `EAGER` pipeline. This allowed us to compare our reconstruction of the two genes to the reconstruction generated by the standard method using the full genome as a reference. To identify specific variations in either copy of the genes, we searched for heterozygous positions in the individual reconstructions of the extracted sequences that show an allele frequency between 25% and 75%.

Finally, we applied the full `DACCOR` pipeline to reconstruct the whole genomes, including the identified repetitive regions, of two syphilis samples. For this we chose two samples from *Arora et al. (2016)*, one with a moderate coverage (*AR1*, 7X, SRR3268681) and one with a very high coverage (*AR2*, 157X, SRR3268682). These two samples were also assembled using the de novo assembler `SPAdes` (version 3.11.1) (*Bankevich et al., 2012*) in order to compare repeat resolution approach to a pure de novo assembly. For this assembly, the reads were adapter clipped and quality trimmed to a minimum Phred-quality of 20, prior to the assembly. The resulting contigs were then mapped against the reference genome with `BWA MEM`, using `MADAM` (*Seitz & Nieselt, 2017*). All bases in the reference genome where contigs mapped were counted as resolved bases and the ones where no contig mapped as unresolved. They were then compared with the results of `EAGER` without repeat resolution and `DACCOR` with repeat resolution.

If not specified differently, all programs were run with default parameters.

## RESULTS

We first evaluated the identification of repetitive regions by comparing these to the ones identified by `VMatch`. This comparison (see Table 2) shows that the results of both programs are almost identical. We considered `VMatch` as the "golden truth" and could, therefore, compute an accuracy for the repetitive regions reported by `DACCOR`. We achieved an excellent accuracy with over 99% in all tested cases with only very few false negatives as well as false positives.

Next, we reran `DACCOR` with a higher sensitivity allowing for up to five mismatches in repetitive regions of lengths at least 101 base pairs. These results of the different bacterial genomes (see Table 3) show that the *S. flexneri* genome contains by far the most repetitive regions (1,249 compared to 29 in *T. pallidum*, 242 in *E. coli*, and 190 in

**Table 2 Comparison of the identified repetitive positions in different bacterial genomes of DACCOR and VMatch, which was seen as the golden truth to evaluate against.**

|  | T. pallidum | S. flexneri | E. coli | M. leprae |
|---|---|---|---|---|
| True positives | 22,382 | 376,669 | 107,456 | 74,406 |
| True negatives | 1,116,478 | 4,211,322 | 4,535,106 | 3,192,194 |
| False positives | 0 | 354 | 463 | 131 |
| False negatives | 773 | 18,857 | 3,307 | 1,472 |
| Accuracy (%) | 99.93 | 99.58 | 99.92 | 99.95 |

Note:
Both programs were run allowing for one mismatch and reporting only regions of at least 101 bp. The $k$-mer size for DACCOR was set to 17.

**Table 3 Statistics of identified repetitive regions using four different bacterial genomes for a $k$-mer size of 17, at most five mismatches, and a minimum length of 101 base pairs.**

|  | T. pallidum | S. flexneri | E. coli | M. leprae |
|---|---|---|---|---|
| Genome size (bp) | 1,139,633 | 4,607,202 | 4,646,332 | 3,268,203 |
| # Repetitive regions | 29 | 1,249 | 242 | 190 |
| Different repetitive regions | 14 | 482 | 104 | 76 |
| Maximum length of repetitive regions | 3,283 | 5,383 | 3,141 | 2,578 |
| Average length | 823 | 570 | 706 | 643 |
| Repetitive regions with mismatches | 1 | 127 | 47 | 9 |
| Sum of repetitive bases | 23,892 | 660,261 | 160,897 | 119,871 |
| % of genome repetitive (non overlapping) | 2.0 | 8.3 | 2.4 | 2.3 |

*M. leprae*). It also contains the longest repetitive region of the four bacteria (5,383 compared to 3,283, 3,141, and 2,578, respectively). The average lengths of the repetitive regions are quite similar in all four bacteria (between 570 and 823 base pairs). The genome of *T. pallidum* contains 29 repetitive regions, of which 14 are different. It is apparent that 13 of those 14 are contained precisely twice whereas one is contained three times. This distinction is not as apparent in the other genomes, where more regions have to be contained more than twice. The number of repetitive regions that can be identified when allowing for up to five mismatches also varies between the different bacteria. There are 127 of regions containing mismatches in the *S. flexneri* genome, whereas there are only 47 in the genome of *E. coli*, nine in *M. leprae*, and one in *T. pallidum*. Overall 8.3% of the *S. flexneri* genome is comprised of repetitive regions. The genomes of *E. coli* and *M. leprae* are comprised of 2.4% and 2.3% repetitive regions, respectively, and only 2.0% of the *T. pallidum* genome is repetitive.

Next we assessed the runtime of DACCOR, which was calculated on a server with four *Intel(R) Xeon(R) CPU E5-4610 v2 @ 2.30GHz* and 500 GB of memory with the GNU time command. The runtime linearly correlates with the number of identified repetitive bases (see Fig. 2). After an initial preprocessing step, DACCOR identifies about 14,000 repetitive bases per minute (using one mismatch).

The runtime of DACCOR correlates mainly with the repetitiveness of the genome (see Fig. 2), i.e., with the number of repetitive bases that are identified in the respective

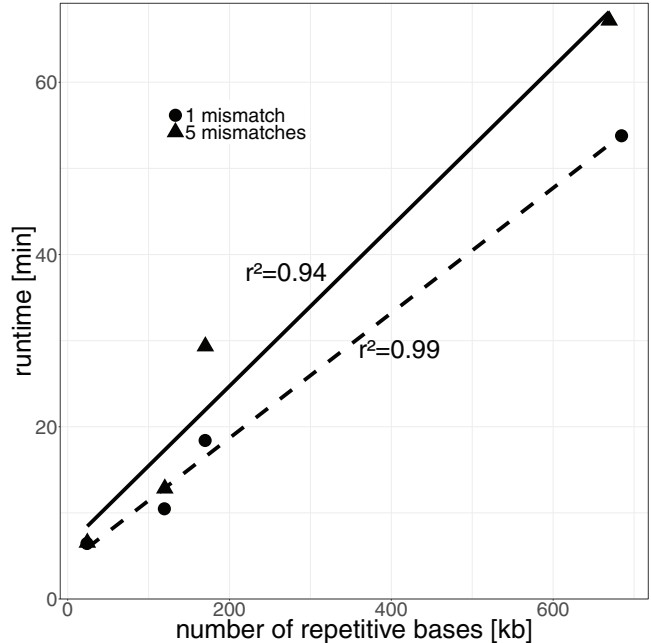

**Figure 2 Runtime of DACCOR in relation to the number of identified repetitive bases.** The dashed line represents the regression for one mismatch, the solid line for five mismatches.

genome. When allowing more mismatches, a slight increase in runtime is also noted. The $k$-mer size does not influence the runtime (see Fig. S2). We compared the runtime of DACCOR with Vmatch. DACCOR identifies repetitive regions with several mismatches faster than VMatch, however for repetitive regions with few mismatches, VMatch is faster (see Fig. S3).

When analyzing the results of DACCOR on the genome of *T. pallidum*, the two longest identified repetitive regions in the *T. pallidum* genome correspond to the 16S and 23S rRNAs, respectively. Thus one region contains one operon of a gene. We extracted these regions and used them as two independent references for their reconstruction of all samples published by *Arora et al. (2016)*, *Pinto et al. (2016)*, and *Sun et al. (2016)*. We mapped all of the sequencing reads against each copy of the gene individually, reconstructed them for each sample, and counted the number of resolved bases in each copy. This number was compared to the number of resolved bases in the two copies of the respective gene when mapping against the whole genome without repeat resolution. We then computed the difference between these two numbers and divided by the length of the gene. With our approach, we could improve the base pair resolution (see Fig. 3) by a median value of 82.7% for the 16S and 87.4% for the 23S rRNA (see Fig. S4 for a detailed plot for each sample). It shows that the percentage of the resolved base pairs was at least as high when mapping only against the extracted sequences, compared to the mapping against the whole genome for all analyzed samples. This means that we do not lose resolution when mapping only against these sequences. On the other hand, 99 out of 106 and 103 out of 106 samples in the 16S and 23S rRNA, respectively, gain

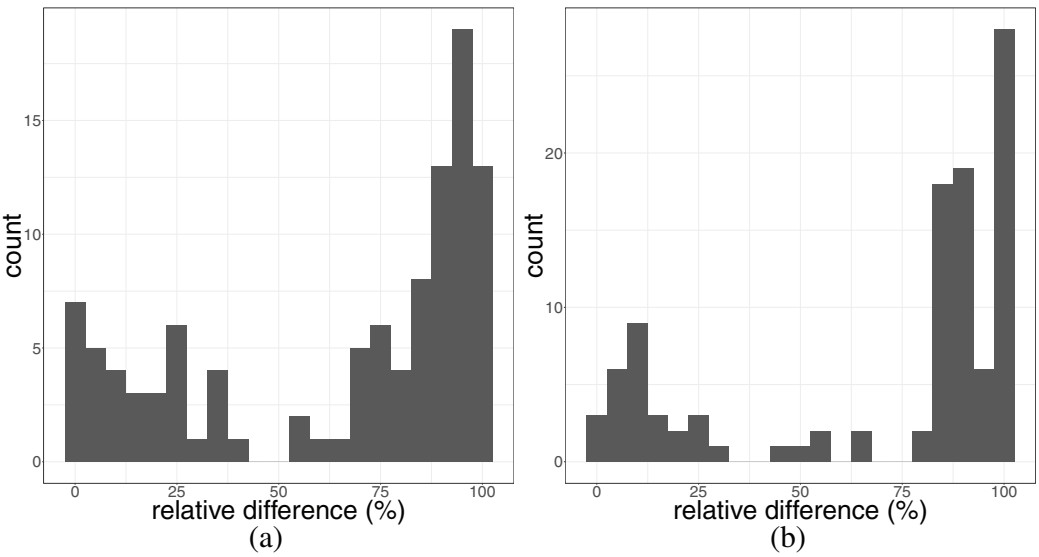

**Figure 3 Histogram of resolved bases (as fraction relative to the length of the respective gene) of the 16S rRNA (A) and 23S rRNA (B) in 106 clinical syphilis samples (*Arora et al., 2016*; *Pinto et al., 2016*; *Sun et al., 2016*).** The number of resolved base pairs, when mapping against each copy of the gene individually in comparison to the standard mapping-based approach using both copies, has been computed.

**Table 4 SNPs in the 16S and 23S rRNAs of *T. pallidum* identified after extracting the repetitive region.**

|  | 16S rRNA | 23S rRNA |
| --- | --- | --- |
| Length of gene | 1,495 | 2,900 |
| Number of variant positions | 36 | 46 |
| Number of site-specific positions | 30 | 43 |

Note:
The *number of variant positions* is the number of all positions where at least one sample shows some kind of genetic variation. Out of those positions, the *site-specific positions* refer to variant positions that appear to be different in the two copies of the respective gene (allele frequency between 25 and 75%) in at least one sample.

information up to an improvement of 100% of the sequence of the respective gene. In these most extreme cases, not a single base could be reconstructed without DACCOR, whereas with it 100% of the gene could be reconstructed.

Using the individual reconstructions of the 16S rRNA and the 23S rRNA gene sequences, we also tried to identify SNVs or SNPs that are specific for only one copy of the respective gene. Hereafter we refer to these variations using the term SNP, though we cannot confirm that the variation is present to a minimum degree within a population (e.g., ≥1%). The results of this analysis identified 36 positions that have a SNP call in at least one of the 106 samples for the 16S rRNA (see Table 4). Of those 36 positions, 30 show evidence for a site-specific SNP in at least one of all analyzed clinical samples. There are two positions (884 and 888 relative to the start of the 16S rRNA), which show site-specific variance in about 20% of the samples (see Tables S1 and S2). In the 23S rRNA, there are 46 positions where at least one sample has a SNP. Of these 46, 43 show evidence for site-specific SNPs in at least one sample. Here, one position (2,003)

**Table 5 Statistics for the assembly with SPAdes of the two clinical syphilis samples (*AR1* and *AR2* from *Arora et al. (2016)*).**

| Sample | # contigs | N50 | Average contig length | ≥1,000 | ≥10,000 | Longest |
|--------|-----------|-----|------------------------|--------|---------|---------|
| AR1 | 43,162 | 247 | 245.9 | 396 | 1 | 10,286 |
| AR1 mapped | 1,137 | 978 | 696.61 | 174 | 1 | 10,286 |
| AR1 filtered | 396 | 1,941 | 1,895.61 | 396 | 1 | 10,286 |
| AR1 filtered mapped | 174 | 2,433 | 2,235.41 | 174 | 1 | 10,286 |
| AR2 | 278,193 | 278 | 254.02 | 1,589 | 38 | 75,865 |
| AR2 mapped | 852 | 17,810 | 1,527.99 | 105 | 38 | 75,865 |
| AR2 filtered | 1,589 | 1,670 | 1,981.73 | 1,589 | 38 | 75,865 |
| AR2 filtered mapped | 105 | 20,441 | 10,693.95 | 105 | 38 | 75,865 |

Note:
The *sample* column specifies the postprocessing step; "*filtered*" refers to contigs of 1,000 bp or more and "*mapped*" are only those contigs that could be mapped against the reference genome of *T. pallidum*.

**Table 6 Enhanced genome resolution of two clinical syphilis samples (*AR1* and *AR2* from *Arora et al. (2016)*).**

| Sample | Mean coverage | Method | #N | %N |
|--------|---------------|--------|-----|-----|
| AR1 | 7X | EAGER | 23,348 | 76.98 |
| | | SPAdes | 12,459 | 41.08 |
| | | SPAdes filtered | 17,336 | 57.16 |
| | | DACCOR | 4,473 | 14.75 |
| AR2 | 157X | EAGER | 17,549 | 57.86 |
| | | SPAdes | 9,247 | 30.49 |
| | | SPAdes filtered | 14,858 | 48.99 |
| | | DACCOR | 964 | 3.18 |

Note:
EAGER indicate the results using only the full genome as a reference without the extra repeat resolution of DACCOR and SPADES indicate the results of the de novo assembly. Here "*filtered*" stands for the results using only contigs of length ≥1,000 bp. The values refer to the repetitive regions only, including the margin regions (in total 30,330 bp).

shows evidence for site-specificity in 37% of the samples. Additionally, nine positions show site-specificity in at least 15% of the *T. pallidum* samples.

Finally, we compared the mapping-based approach that is used by DACCOR and EAGER with the de novo assembly approach using SPAdes. For this, we used two clinical syphilis samples (*AR1* and *AR2*), which had medium and high genome resolution in the study of *Arora et al. (2016)*. Table 5 shows standard assembly statistics using various postprocessing methods on the contigs of the two samples. For the comparison of SPAdes with EAGER and DACCOR (see Table 6), we used all contigs of SPAdes that mapped against the reference genome. For the medium covered sample *AR1*, 85.3% of the repetitive regions could be resolved, compared to the 58.8% with SPAdes, and 23.0% using EAGER without DACCOR. For the high coverage sample *AR2*, 96.8% of the repetitive bases could be resolved, compared to the 42.1% using the standard mapping-based approach and 69.5% with the de novo assembly. Using only contigs greater or equal to 1,000 bp resulted in less resolved bases in the repetitive regions.

## DISCUSSION AND CONCLUSION

We have developed DACCOR, an approach to increase the base pair resolution of repetitive regions during the mapping-based reconstruction of full genomes using short reads. For this, we first identify the repetitive regions of a given reference genome de novo. These regions are then used as individual reference sequences for the mapping of short reads of NGS samples. Finally, a new draft genome is created by combining the reconstructed repetitive regions with the rest of the genome that has been reconstructed using a standard mapping-based approach.

Our de novo identification of repetitive regions uses a $k$-mer based approach. The choice of $k$ is important for the identification of repetitive regions. For de novo assembly based on *de Bruijn* graphs, the optimum value for $k$ depends on the genome length, the coverage, the quality, and the length of the reads (*Zerbino & Birney, 2008*). In the case of different read lengths, as often observed in aDNA projects, it has been shown that using multiple different $k$-mers improves the assembly (*Seitz & Nieselt, 2017*). Other approaches choose an optimal $k$-mer size so that the uniqueness is maximized (*Gardner & Hall, 2013*). However, as we want to identify repetitive regions, we do not want to choose a $k$-mer size to maximize uniqueness. In our approach, the $k$-mer size defines the minimum length of the repetitive regions that can be identified in the first step. Thus, a $k$-mer size as small as possible should be used to be able to identify all putative repetitive regions. However, very small $k$-mer sizes lead to an exponential increase in the runtime, due to the increasing number of random occurrences of these small $k$-mers that have to be accounted for. On the other hand, the $k$-mer size should also not be longer than half the length of the input reads, as repetitive regions with one mismatch of the same length as the reads could otherwise not be identified. Therefore, we propose a minimum size for $k$ of 17.

To identify all repetitive $k$-mers, all possible $k$-mers are stored in the first step of the identify subprogram of DACCOR. Those that are not repetitive are removed after the screening. Nonetheless, this results in high memory usage if the genome contains many repetitive regions. For the example of *S. flexneri*, we observed a memory footprint of 8 GB with a $k$-mer size of 17.

The comparison between DACCOR and VMatch showed that VMatch is slightly more sensitive, probably due to its suffix array approach. A possibility to increase the sensitivity of DACCOR would be to elongate all identified repetitive regions based on a local alignment. In principle, DACCOR could also use Vmatch, as well as other repeat finding methods, to replace our de novo repeat finder. However, to automate this, further adaptions would be necessary.

When mapping short reads from typical NGS data, a number of approaches recommend to use genomes as references whose repetitive regions are masked (*Tarailo-Graovac & Chen, 2009*). However, this may be problematic because repetitive regions often overlap and can be quite complex. An example for this is the *arp* gene of *T. pallidum* (see Fig. 4). It contains several overlapping repetitive regions. It can be seen that the region labeled *repeat_3* is partly repetitive with itself. Thus a masking of one of the repetitive regions would mask most of itself. Additionally, the masking of the second occurrence of

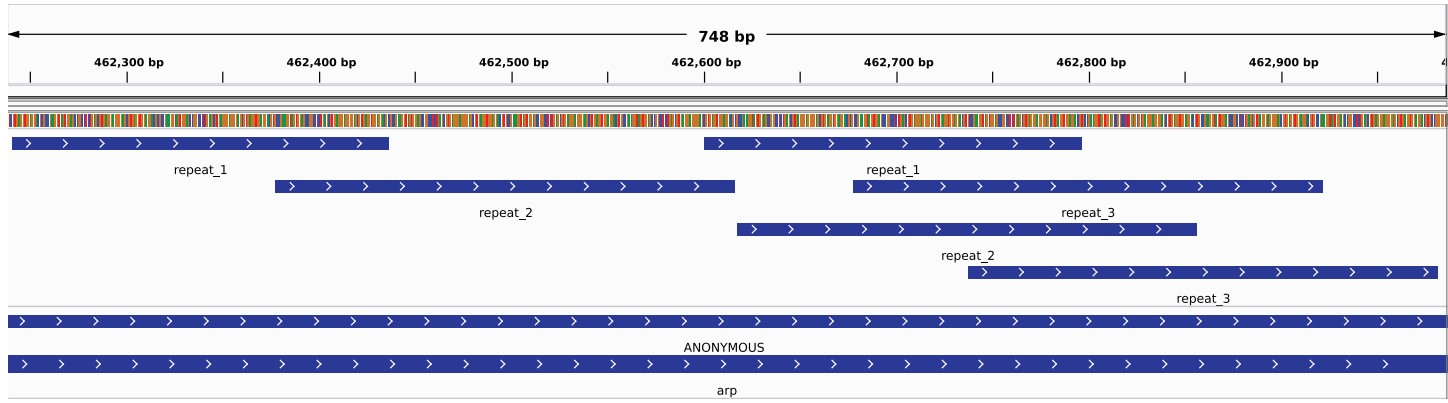

**Figure 4** Repetitive regions contained in the *arp* gene in *T. pallidum*, de novo identified by DACCOR with a *k*-mer size of 17 and no mismatches.

*repeat_1* would also mask most of both occurrences of *repeat_3*. When masking the first occurrence of *repeat_1*, the masking of either occurrence of *repeat_2* would also mask part of the unmasked region of *repeat_1*. Thus, masking of repetitive regions could result in either losing genome information or leaving repetitive regions unmasked. We, therefore, propose to use the identified individual repetitive regions as separate references for the mapping and merge all individually reconstructed regions into a common draft genome.

Since the de novo identification of repetitive regions in genomes, using these then as individual references for the mapping and merging all reconstructed genomic regions into a final draft genome requires many different steps, our primary goal of DACCOR was to present an entirely automatic procedure encompassing all these steps. DACCOR makes use of EAGER, a pipeline for the automatic reconstruction of genomic data sets. For highly identical repetitive regions, each copy is stored as individual sequences together with a margin at the 5′ and 3′ end of the region. We propose to set the margin to the length of the longest read that will be used in the mapping.

Using DACCOR, we have shown that a higher base pair resolution in repetitive regions, compared to the reconstruction using the standard mapping-based approach, can be achieved. In the standard approach, reads that can be mapped to different locations result in a mapping quality of zero, which in turn decreases the genotyping quality (*McKenna et al., 2010*). Additionally, our results show that de novo assembly also cannot reconstruct all repetitive regions completely, even though the results are better than the standard mapping-based approach without repeat resolution. By mapping to the repetitive region only, these reads have a higher mapping quality, a higher genotype quality, and thus result in more resolved positions. However, one has to acknowledge that other unresolved bases that may stem from a lack of coverage or low sequencing quality, cannot be resolved with our approach. In the case of the two syphilis samples, we could show that the majority of unresolved bases in repetitive regions could be resolved. The remaining unresolved bases are either outside of the identified repetitive regions or without read support within the regions. Furthermore, we have shown that

using `DACCOR`, the identification of SNPs in repetitive regions can be improved. This is especially useful for the 23S rRNA, as it is known to play a role in the antibiotics resistance of this bacterium (*Arora et al., 2016*).

In conclusion, we have developed an entirely automatic pipeline that first conducts a de novo repeat identification in bacterial genomes and then uses the repetitive regions for an enhanced mapping of short read NGS data. Increasing the resolution of a draft genome affects many downstream analyses, such as population genetics or phylogenetic analyses. For future improvements, we plan to reduce the runtime and memory usage by adjusting our data structure and by adding more parallelization to some of the computing steps. With this, we hope to eventually be able to identify repetitive regions also in large eukaryotic genomes, like the human genome.

### Funding

This work was supported by the Deutsche Forschungsgemeinschaft and the Open Access Publishing Fund of University of Tübingen. The funders had no role in study design, data collection and analysis, decision to publish, or preparation of the manuscript.

### Grant Disclosures

The following grant information was disclosed by the authors:
Deutsche Forschungsgemeinschaft.
University of Tübingen.

### Competing Interests

Kay Nieselt is an Academic Editor for PeerJ.

### Author Contributions

- Alexander Seitz conceived and designed the experiments, performed the experiments, analyzed the data, contributed reagents/materials/analysis tools, prepared figures and/or tables, authored or reviewed drafts of the paper, approved the final draft.
- Friederike Hanssen authored or reviewed drafts of the paper, approved the final draft.
- Kay Nieselt authored or reviewed drafts of the paper, approved the final draft.

### Data Availability

We have developed an automated software pipeline, written in Java, which allows other researchers to use our methodology. This pipeline is available on GitHub: `DACCOR` https://github.com/Integrative-Transcriptomics/Daccor.

### Supplemental Information

Supplemental information for this article can be found online at http://dx.doi.org/10.7717/peerj.4742#supplemental-information.

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
