# Peer review of "DACCOR–Detection, characterization, and reconstruction of repetitive regions in bacterial genomes"

_PeerJ, doi:10.7717/peerj.4742_

## Round 0.1 · original submission · Major Revisions

Dear Alexander,

We now have the three review reports back. All of the reports indicate your work is interesting and can add value to the scientific community. However, there are a number of issues in which you and colleagues need to spend time and effort to sort out. Details are attached.

I look forward to your responses and a new version of the paper!

Best regards,

Zemin

Reviewer 1 ·

Basic reporting

The manuscript is very well written. Just a couple of typos:

1. Table 1. Commas missing in false negatives row.

2. Table 2. Max length of repetitive regions for E. coli should have a comma instead of period.

Experimental design

3. De novo assembly is almost completely ignored throughout the manuscript, but is one obvious way of reconstructing a genome sequence from reads. This should be discussed because it is the only viable option where a reference genome is not available, or where one cannot be sure that the reference is close enough to the reads to use a reference-based method. For example, “any one E. coli contains less than 10 % of the total number of E. coli genes in the E. coli pan-genome” (https://www.ncbi.nlm.nih.gov/pubmed/25722247).

I think it is important to compare DACCOR against a state of the art assembler, ideally SPAdes because it is currently the most popular for bacteria genomes. Show how well SPAdes reconstructs the rRNA genes compared to DACCOR and EAGER, and add the results to the end of the results section on page 6. If DACCOR is as good as advertised, then it should do better than SPAdes.

4. A few details need adding for reproducibility:
* All accession IDs for the genomes and reads used.
* Version numbers of all software used.
* Options used for all software - if it was the defaults then still state it.
* Any scripts written to help produce the results should be released, eg on github.
* How was the run time measured and on what type of machine?

5. Can the authors justify why it is ok to take VMatch as the truth (lines 144-148)? Is it guaranteed to find every repeat?

6. I tried out DACCOR and VMatch on a toy made up FASTA file that had repeats with a few SNPs and indel differences. VMatch can handle indels in its matches, but it appears DACCOR cannot. Repeats reported by DACCOR were broken where I put an indel, eg two copies of a sequence with the only difference being a single nucleotide indel in the middle. Ultimately I don’t think this matters to the overall method of DACCOR and the quality of the results, but could it explain some of the false negatives in Table 1? In any case, I think it would be good to clarify that allowing “mismatches” when running DACCOR really means SNPs only, not indels.

Validity of the findings

No comment

Additional comments

The manuscript is clearly written, and DACCOR looks like a useful tool for the community. The software looks well engineered and even appears to have unit tests.

My only other comments relate to the software:

7. Make a versioned release on github, corresponding to the current version (0.0.1) of the code.

8. Include the JAR file in each release, to save the user compiling (eg like Picard has). I had issues compiling on Ubuntu Zesty. gradle build returned the error:
java: symbol lookup error: /usr/lib/jni/libnative-platform-curses.so: undefined symbol: tgetent

I expect the cause was this known bug:
https://bugs.launchpad.net/ubuntu/+source/libnative-platform-java/+bug/1683761
but I didn’t pursue it further.

To be clear: this is not a fault with the DACCOR code, but a problem with that version of Ubuntu. However, releasing a JAR file should solve this.

I had no problems compiling the software on Ubuntu Xenial. I did not test on a Mac (or WIndows!).

Reviewer 2 ·

Basic reporting

The manuscript is well written and organized. However, the introduction is sometimes difficult to follow for people not coming from the bacterial genome field and the subject and the aim of the project after the first read of the introduction is not entirely clear. With respect to the wide scope of the journal this needs to be improved. The description of the method to reconstruct repetitive regions, the results and the discussion are generally of good quality but need at some points improvements (see below in the section “Comments for the author”). The figures are relevant to the content of the text and understandable.

Experimental design

No comment here. Please see line-by-line remarks in the section “Comments for the author”.

Validity of the findings

No comment here. Please see line-by-line remarks in the section “Comments for the author”.

Additional comments

1) Introduction

Line 23
The term genome reconstruction needs to be clearly defined. From the context it seems to me like reference based genome assembly but I might be wrong. It is also highly recommended to make clear, that this is all in the field of bacterial genomes at a very early point. Why do you pick especially BWA from the dozens of mappers? Is it necessary to this very one.

33ff
The point of the last two sentences of the first paragraph is unclear. Filtering low quality mappings is no problem – it is a unavoidable step.

37
Please change “role in the genome” to “role in many genomes”

40f
The author’s statement is not the same as said in the reference (Denœud & Vergnaud, 2004) which reads “The frequent observation that tandem repeat-containing genes are often associated with outer membrane proteins suggests that such genes help bacteria adapt to their environment, …” . I just want to point out that TR regions do not encode proteins.

51f
Please rephrase the sentence. ”Sequencing … result in short DNA fragments” makes no sense. Sequencing results in reads.

59ff
The overview of de novo repeat identification methods should be extended. It is not sufficient to only cite the 15 year old VMatch. There are many other strategies out there.

63ff
This final paragraph is not completely clear to me and should be rephrased. Please also add at the respective sentence that NGS data was used.

Figure 1
The caption talks about 6 steps and nothing else. Please extend/adapt. It would be good to adapt the right hand side explanations to the style of the left box. A general aesthetic refinement of this figure is highly recommended.
If DACCOR is that modular as said in lines 86-89, it would be good to indicate this somewhere near the “identify repeats de novo”-box.


2) Methods

78f
The explanation of Step 3 is not clear to me.

80
Would it be possible to explain this mismatch identification in a kind of sketch? Especially the mismatch-marker addition sounds unique to me and is maybe worth a figure.

90/95
“repeat/repetitive region”: Please select one term and use it throughout the entire manuscript uniformly. I prefer “repetitive”.

100
“sequenced fastq files” sounds odd. Fastq files are generated during sequencing.

104
Why is BEST capitalized?

109-111
The name of this module could be given much earlier. Consider moving this tiny paragraph after line 85.
(minor remark: the name of this module is called “analyze” in the corresponding github README.md file)

114ff
This is too much of the term “reconstruction” and confuses the reader a lot. Please define other terms or at least rephrase this paragraph. For example, does “reconstructed subsequences” mean “reconstructed repeats”? What is the difference between “reconstructed regions” and “whole genome”?

119-123
This paragraph is pretty clear. It is also worth to consider putting the module names into the respective boxes of Figure 1.

128
Please avoid “several”, just give the number. This is also a good place where the species are named once with their full name. Then please stick to the scheme to give at first appearance the full name and afterwards always in the style “E. coli”. Please avoid strain names (Nichols) if possible. Also the references to the bacterial genomes are missing (at least provide an accession number).

132
provide the number of samples (106?)


3) Results

144-148
This is contrary to line 130 where it is stated to use a max of 5. A max of 5 is not used for the VMatch comparison but only for the DACCOR-only repeat identification resulting in Table 2, right?

Table 2
replace “.” by “,” in 3.141. Please avoid “several”. There are also different spellings of “base pairs” throughout the manuscript (with and without dash).

162
The wrong supplemental figure is referenced (also in line 209)

Table 3
remove italics in caption

174f
This last statement is not clear to me. What is “almost all” and where does the “100%” come from?

181
Please provide the exact material reference (like “Supplemental Table 1&2”). I guess it is referring to the two csv files. They are actually not comma-separated but tab-separated. In addition, the 16S.csv lacks headers in the last columns. Although this header is provided in the 23S.csv, it is still not very descriptive and needs to be improved.

Figure 3
The labels (a)&(b) need to be resized


4) Discussion

213f
I assume this refers to Supplemental Figure 1, which is not referenced throughout the entire manuscript. Please refer to this comparison between DACCOR and VMatch in the results and discussion sections appropriately.

238f
Why twice max read length? It seems too long to me as reads can then be mapped without “touching” the repetitive region. If this is intended, then please explain it in the manuscript.


Generally, I support this article and encourage the authors to submit a revised version to allow a re-evaluation as I belife this automated pipeline for repeat reconstruction can be of interest to several fields of genomics.

Reviewer 3 ·

Basic reporting

The authors propose an approach for reconstructing repetitive regions in bacterial genomes. There are several major issues and minor issues:

Major issues:
1. The authors claim their method DACCOR a novel method for reconstructing the repetitive regions. The first step (identify repeats de novo step) seems the only novelty step, as other steps more like pipeline that heavily relies on EAGER. However, the authors only use one paragraph (line 74-85) to describe this novolty part. Most of the descriptions are unclear, for example:
1) In the "Merge k-mers" step, how the authors deal with circle or bubbles, and process branches?
2) How the "Add mismatch marker" step goes? How can this allow lots of mismatches like 15 in supplementary figure 1? How to deal with indels?
3) What is the "Filtering" step? No any description here.
Besides, for doing comparison, many other tools can also do the similar work efficiently and accuratly, for example RepeatScout, the authors need to prove why their approach DACCOR is better?

2. The authors compare the performance of the first step (identify repeats de novo step) against VMATCH in table 1, and use the VMATCH results as benchmark. Then, how can the authors prove their approach is better? If not as good as VMATCH, then why users prefer the authors' approach? In supplementary figure 1, the authors showed that their approach DACCOR is faster than VMATCH, especially when number of mismatches is large, but how accuracy of their approach? The results and comparison are not promising.

3. There are lots of available long reads can be used as benchmark data for comparison. The authors should provides comparison with promising benchmark data.

Minor issues:
1. line 298-332: the authors use half page to list the authors of a paper in reference! But only one paragraph for describing their major part of Method!!
2. The language shoule be improved, for example:
1) line 75: the ... is divided into ... that occur more than once are stored.
2) line 78: long regions consisting only of one base?
3) line 139: 25 -> 25%

Experimental design

no comment

Validity of the findings

no comment

Additional comments

no comment

External reviews were received for this submission. These reviews were used by the Editor when they made their decision, and can be downloaded below.

---

## Round 0.2 · Minor Revisions

Dear Alexander,

We have two review reports back and they both think the manuscript has been improved significantly, compared with the previous version. Both reviewers still have some issues and concerns. Based on the two reports, I suggest:

(1) Following review 2's suggestion, make a new figure about mismatch identification and put it in the supplemental file;

(2) Make all the small changes as suggested by the two reviewers.

I look forward to receiving your new version in due course.

Best regards,

Zemin

Reviewer 2 ·

Basic reporting

No comment here. Please see line-by-line remarks in the section “Comments for the author”.

Experimental design

No comment here. Please see line-by-line remarks in the section “Comments for the author”.

Validity of the findings

No comment here. Please see line-by-line remarks in the section “Comments for the author”.

Additional comments

The Authors addressed all my comments very well by either changing the manuscript or via a comment. This improved the manuscript remarkably but a few minor issues and a more serious one remain to be solved.

Minor:
L13: The part “of the genotypers” can be removed.
L24: Naming SPAdes this way in the introduction is not needed.
L73: Both de novo tools have the same reference.
L111: Would it make sense to add “separately” in this sentence to underpin the usefulness of DACCOR? “… and can be used SEPARATELY for the de novo …”
L116f: For me the sentence “Additionally…” looks redundant. If this is not a “special feature” (which I then do not understand) it can be deleted.
L133-135: This paragraph feels redundant or at the wrong place. I think it can be deleted.
L143-147: This also repeats basically the lines written above. If the authors want to make the point, that “pipeline” connects the three modules, they should write this in a shorter/better way.
L148-151: Wouldn’t it make sense to put this paragraph right to the other one above that talks about “identify”?
Figure 1: The figure improved a lot. My only suggestion refers to the style of the detail boxes. I would prefer that they have the same grey background like the center boxes and have the included sub-boxed with a white background. This would emphasize that they are a detailed explanation of the three center boxes. This effect can even made be stronger by replacing the curly brackets by dashed lines between the corresponding corners of the boxes (a bit like in [1]). The “incoming” and “outgoing” arrows in the detail boxes should also be avoided as they are redundant with the arrows in the center column.
[1] https://www.researchgate.net/figure/Diagrammatic-cross-section-of-human-skin-including-a-zoomed-in-view-of-the-epidermis_fig1_321341598
Table 1: This is a very good overview. Please substitute the word “read-seq data” by something like “DNA-seq data” or “NGS data”. Also the number of samples coming from the 3 studies should be indicated either in this table or near L158 in the text. Simply to explain how they sum up to 106.
L156: The reference for VMatch was already given in L69.
Table 3: “different repetitive regions” needs to be explained in the text or rephrased in the table. I don’t understand what it is supposed to mean. Is it something like a repeat family?
Table 4: It should be indicated that this data refers to T. pallidum. Just out of curiosity, are the two copies of the 16S and 23S exactly the same length, because only the two numbers 1495 & 2900 are given. By the way, the thousands separators are missing.
L206-209: It should be indicated that the two longest regions contain both copies of the respective genes (1 region = 1 operon).
L223-232&L302: I’m not a variation specialist, but was always told “be careful with the term SNP as it refers to populations”. I suggest to replace SNP by a term like “site-specific / variant position” like it is already used in Table 4 and the Methods.
L225: Am I getting it correctly that these 36 positions are referring to the two 16S copies together? This means in turn that besides the 30 site-specific variations 6 are in common on both copies? If yes, this should be formulated a bit clearer, if no, it needs even more clarification.
L230: Change 2003 to 2,003.
L242: Change 1000 to 1,000.
L250: Set de novo in italics.
L256: “use a k-mer size” sounds odd. Maybe “choose” or “optimise”?
L261: “On the other hand, the k-mer size should also not be longer than half the length of the input reads” It should be explained why this is suggested.
L271: A period is missing.
Table 5: Thousands separators in the last column are missing.
Table 6: The sings in column “%N” are not consistent. The table could also be improved by either replacing the “%N” column by “% resolved bases” or adding a new column with these numbers. This is only a suggestion.
Figure 4: The caption should be improved with some additional information like starting with “Repetitive regions de novo identified by DACCOR…”.

General remarks:
Some Program names were not yet converted to Courier like RepeatMasker, BWA or GATK.
I recommend to change all occurrences of “mapping approach” to “mapping-based approach” throughout the manuscript. (for example in L241, 249, 296)
Although I found several typos, the manuscript should be checked again very carefully. For the next review-round I am expecting a much higher quality in this regard.


Major:
My original comment:
Line 80
Would it be possible to explain this mismatch identification in a kind of sketch? Especially the mismatch-marker addition sounds unique to me and is maybe worth a figure.

The Authors’ reply:
We explained it in more detail. However, we feel that if we add a figure, the focus will shift too much to this repeat finding approach and away from the actual reconstruction of repetitive regions, which is our main focus. The repeat finding itself is an integral part of DACCOR as we would like to publish it, but as we explained, it is not the main contribution and can be substituted.

My reply:
Although this paragraph was extended a lot, I am still not totally sure about this mismatch-marking. I do not agree with the authors’ reply. If the method is novel and part of that publication, it has to be explained in an adequate and understandable way. It is a methods focused paper. However, I understand that a figure in the main text really might shift the impression of the reader. Therefore, my suggestion is to make a supplemental figure.

Reviewer 3 ·

Basic reporting

no comment

Experimental design

no comment

Validity of the findings

The authors have resolved many issues for my previous review, and the draft has been improved a lot. However, there are still one major and one small issues:
1) Major:
The major contribution for the paper is they propose a pipeline that can reconstruct the genomes, and also the repeats for individuals. However, it is hard to evaluate the accuracy of the constructed repeats and genomes. The authors have tried several indirect ways to "prove" their pipeline is accuracy, at least not worse than other tools. For example, they did statistic comparison at line 219-235, and also compared with de novo assembly results. The authors argue that long reads is not as common as NGS data, which is true. But they ignore that long reads can be used as a good resource for benchmarking their approach. And there are far more data can be used. For example, data released here: http://www.cbcb.umd.edu/software/pbcr/closure/index.html (there are even error corrected long reads can be used). So, it will be much more convincing if the authors can use long reads (and the assembly) as benchmark to evaluate their pipeline performance.

2) Minor:
I think this is the right way (authors should adjust according to PeerJ requirement) to cite this paper, not take half a page to list all the names:

International Human Genome Sequencing Consortium. "Initial sequencing and analysis of the human genome." Nature 409.6822 (2001): 860.

Additional comments

no comment

---

## Round 0.3 · accepted · Accept

Dear Allexander,

Based on the reports from all the reviewers, I am pleased to let you know that your manuscript has been accepted for publication. Also there are small issues raised from a reviewer and I hope you can address them with your final version while in production.

I take this opportunity to thank you for publication with us and truly looking forward to working with you again in the near future!

Best regards,

Zemin

# Reviewer 2 ·

Basic reporting

Please see "General comments for the author"

Experimental design

Please see "General comments for the author"

Validity of the findings

Please see "General comments for the author"

Additional comments

The Authors addressed my comments very well. I especially like the improvements in Figure 1 and appreciate the creation of Supplemental Figure 1.

Only 2 minor issues remain.

1) The SNV vs. SNP: I think my intention was mis-understood. I wanted to say, that the term SNV should be used instead of SNP because no “population-wide” frequency can be determined. SNP is often defined as a polymorphic locus with multiple alleles having a certain frequency (often >1%) in a population.

2) Supplemental Figure 1: This figure helps a lot with understanding the mis-match handling. I only stumbled over the equation in Step 5. It is not for all numbers intuitively clear to what they refer to (for example “-2” and “-12”).

Reviewer 3 ·

Basic reporting

The authors have revised the required revisions.

Experimental design

no comment

Validity of the findings

no comment

Additional comments

The authors have revised the required revisions. And hope they can do the validation part with long reads in their future work.